# Comparison of Doppler Ultrasound and Computerized Tomographic Angiography in Evaluation of Cervical Arteries Stenosis in Stroke Patients, a Retrospective Single-Center Study

**DOI:** 10.3390/diagnostics13030459

**Published:** 2023-01-26

**Authors:** Naaem Simaan, Tamer Jubeh, Karine Beiruti Wiegler, Adi Sharabi-Nov, Asaf Honig, Radi Shahien

**Affiliations:** 1Department of Neurology, Ziv Medical Center, Safed 1311001, Israel; 2Faculty of Medicine, Bar-Ilan University, Safed 1311502, Israel; 3Research Wing, Ziv Medical Center, Safed 1311001, Israel; 4Statistics Department, Tel-Hai Academic College, Qiryat Shemona 1220800, Israel; 5Departments of Neurology, Hadassah-Hebrew University Medical Center, Jerusalem 9112102, Israel

**Keywords:** stroke, CTA, DUS, cervical artery, stenosis

## Abstract

There are different diagnostic modalities to investigate atherosclerosis cervical artery disease in suspected stroke patients. We aimed to test the concordance of findings of the two most widely used diagnostic modalities in stroke patients: duplex ultrasound (DUS) and computerized tomographic angiography (CTA). A total of 100 stroke patients were retrospectively included in the study, all of them had DUS followed by CTA. Discrepancies of DUS compared to the CTA results in both the internal carotid and vertebral arteries were found in 44% of the patients. The patients with significant differences in diagnostic results were characterized by older age. Evaluation of the degree of carotid artery stenosis revealed vast differences in patients with 50–69% stenosis found by DUS, in which 45.5% of them had a different percentage of stenosis found by CTA. In studying the degree of stenosis of the vertebral artery, only 47.1% of the patients with more than 50% stenosis found by DUS had the same results with CTA, while the remaining revealed normal or less than 50% stenosis by CTA. The current study emphasizes that CTA is more accurate than DUS in the evaluation of stenosis of the cervical arteries including both the internal carotid and vertebral arteries.

## 1. Introduction

Ischemic stroke can be classified into different categories based on the pathophysiological mechanisms of the focal brain injury, the clinical characteristics, the localization, and type of infarction [1]. The TOAST stroke subtype classification defines five subtypes of ischemic stroke: large-artery atherosclerosis, which may be extracranial or intracranial, embolism from a cardiac source; small-vessel occlusion or lacunar stroke; stroke of other determined etiology; and stroke of undetermined cause [2]. Large-artery atherosclerosis is an important cause of ischemic stroke that carries a higher risk of early recurrent ischemia than other stroke subtypes [3]. Recurrent ischemic stroke remains a challenge for clinicians, even with improved neuroimaging techniques, acute stroke management, and optimal pharmacologic secondary prophylaxis initiated immediately following stroke [4]. Early management of patients with transient ischemic attack (TIA) or stroke is particularly crucial to improve prevention strategies and reduce the ever increasing burden of stroke. 

Most stroke prevention treatments such as cholesterol lowering agents and antiplatelet treatment are unlikely to have an immediate effect or may be unsafe in some patients if implemented too quickly. However, in extracranial internal carotid artery stenosis (ICA), large studies have demonstrated that prompt endarterectomy markedly reduces the risk of recurrent stroke, supporting the need for urgent carotid imaging [5,6]. 

Rapid and accurate diagnostic imaging is critical for stroke management, assessment of prognosis, and treatment evaluation to avoid recurrence. Different diagnostic modalities are commonly used to investigate atherosclerosis cervical artery disease and establish a possible etiology of the patient’s symptoms. These imaging modalities include computerized tomographic angiography (CTA), magnetic resonance angiography (MRA), duplex ultrasound (DUS), and conventional angiography. They have different methods of acquisition but are capable of imaging both the extracranial and intracranial arterial circulation [7].

Computerized tomography angiography (CTA) is a diagnostic modality of choice for extracranial and intracranial vasculature imaging in the clinical setting of acute stroke or TIA [8]. Usually performed following the routine non contrast CT, CTA is a widely available, fast, and non-invasive technique performed with a single bolus of a radiographic contrast agent in the median cubital vein [8,9]. This imaging tool allows us to assess, with high rates of sensitivity, specificity, and accuracy, the status of collateral circulation, injured but potentially salvageable brain penumbra regions, and identify patients who can benefit from reperfusion therapy [10]. The three dimensional vasculature mapping provided by CTA allows quick decisions to be made concerning treatment strategies and the need for neurosurgical interventions. 

MRA is another widely used and noninvasive vessel imaging modality that allows one to assess large vessel occlusions and atherosclerotic lesions in stroke patients [10]. This technique has the advantage of being performed without ionizing radiation exposure and therefore represents a useful alternative for patients with allergies to iodinated contrast material or with renal disease [11]. In comparison to CTA, MRA presents some limitations in acute settings. It is not available in all centers, is significantly more time consuming for image acquisition, and more prone to motion artifacts [10,12]. Furthermore, this imaging modality is contraindicated in patients with claustrophobia, metal implants, or cardiac pacemakers [12]. MRA is limited by flow artifacts, which may result in the overestimation of vessel stenosis, and is inferior to CTA in identifying intracranial structural pathology [10,11].

Doppler ultrasound (DUS) is a relatively cheap, widely available, noninvasive, and safe imaging modality, which has the advantage to be performed directly at the patient’s bedside, allowing for continuous monitoring of the cerebral blood flow velocity [12,13]. It constitutes a method of choice for the screening of carotid artery stenosis and occlusion in patients suspected or at high risk of stroke [12]. TCD is also useful to access collateral circulation to manage cerebrovascular atherosclerotic diseases and detect intracranial vessel abnormalities [8,14]. However, it is highly operator-dependent and less specific and sensitive than other imaging tools such as CTA and MRA [14]. 

Previous studies have compared DUS and CTA results but have focused mainly on the evaluation of carotid artery stenosis (CAD). In these studies, there appeared to be concordance in the results between the DUS and CTA methods. In 1980, Weaver RG et al. compared DUS with arteriography of the carotid bifurcation in 105 patients. The DUS showed false positive results in 19% of tests and false negative results in 56% [15]. Glenn B et al. studied CTA for detection and characterization of carotid artery bifurcation in a cohort of 40 patients who underwent CTA, DUS, and digital subtraction angiography (DSA). This study showed that the correlations between DUS and DSA were poorer than those between CTA and DSA [16]. In 2007, M. Titi et al. compared carotid DUS and CTA in the evaluation of carotid artery stenosis in 107 patients, and showed an overall concordance of 79.1% between DUS and CTA [17]. In 2018, a comparison of the carotid DUS to other angiographic modalities in the measurement of carotid artery stenosis by M. Boyko et al. in 245 patients showed excellent agreement between DUS and CTA [18]. In 2017, DUS for the detection of vertebral artery stenosis was compared with CT angiography in a study by Anouk D. Rozeman. This study showed that DUS has a fair area under the curve for detecting significant stenosis, although adequate assessment of the V1 segment is often not possible due to anatomical difficulties. Assessment of the V2 segment is feasible but yielded few stenosis cases. The usefulness of DUS for the screening of extra cranial vertebral artery stenosis was found to be limited [19].

Most of the studies compared DUS and CTA only in the external ICA with no adequate conclusions, and only one study compared these two modalities in the vertebral arteries and found the DUS modality to be limited. The DUS still represents an easily repeatable test that can be performed in the emergency room as a first-line examination of cervical artery pathology. In our study, we compared between DUS and CTA in the assessment of cervical arteries including carotid and vertebral arteries in the emergency room and at in-patient admission. We consider CTA as the non-invasive gold standard and the most accurate imaging modality for cervical arterial stenosis. We aimed to compare DUS and CTA with regard to its ability to identify stenosis, in order to determine whether DUS alone is a viable imaging alternative in a clinical setting.

## 2. Materials and Methods

The research was performed in accordance with the Declaration of Helsinki and the rules of Good Clinical Practice and received the approval of the ethics committee at Ziv Medical Center (0078-19-ZIV).

This retrospective single center study included 100 acute stroke patients who underwent radiological evaluation that included both DUS and CTA for the four cervical arteries, two carotids, and two vertebral out of 737 patients admitted to our Neurological Department from October 2017 to September 2019. All patients were above 18 years of age, diagnosed with stroke by a senior neurologist, and during assessment in the emergency room and on admission to the Neurology Department. Of the 737 patients admitted to our department with an acute ischemic stroke, only the patients who underwent radiological evaluation with both modalities of DUS and CTA were selected and included in this study. The same US technician performed the DUS on all patients and was blinded to the CTA results. The CTA was performed by the same neuroradiologist who was blinded to the DUS evaluations.

We recorded the patient age, gender, additional demographic data, type of clinical stroke (anterior/posterior), and comorbidities. We additionally recorded the stroke severity according to the four degrees by the National Institutes of Health Stroke Scale (NIHSS) score [20]: mild (<5), moderate (5–14), moderately severe (15–20), and severe (>20). Additionally, the site of stenosis (carotid/vertebral) and degrees of stenosis were recorded. The degree of stenosis determined at gray-scale and Doppler US were stratified into the categories of normal (no stenosis), <50% stenosis, 50–69% stenosis, 70% stenosis to near occlusion, 98% (near occlusion), and total occlusion [21]. We classified the degree of stenosis on CTA in the same way and categorized normal and less than 50% as one group, 50–69%, 70–98%, and near occlusion/occlusion. The results of the CTA and DUS imaging were compared for degrees of cervical artery stenosis.

For the categorical variables, summary tables are provided giving the sample size and frequencies. For continuous variables, summary tables are provided giving the arithmetic mean (M) and standard deviation (SD). Differences between CTA and DUS in terms of proportion were assigned to each of the three group (<50% occlusion, 50–69% occlusion, and 70%+ occlusion) using Pearson’s chi-squared tests. Using CTA as the gold standard, we calculated the sensitivity, specificity, positive and negative predictive values of DUS for measuring the degree of stenosis of right (RT) and left (LT) vertebral arteries. Stenosis was determined according to the NASCET criteria.

Independent sample *t*-tests were applied to measure the age differences between the study groups (with/without difference) and a *p*-value of 5% or less was considered as statistically significant. The data were analyzed using SPSS version 25 (SPSS Inc., Chicago, IL, USA).

## 3. Results

A total of 100 patients (400 arteries) were studied to assess the degree of cervical artery stenosis by both DUS and CTA. The patient characteristics are described in Table 1.

Except for age, we found that none of these factors affected the concordance between DUS and CTA in the evaluation of the degree of stenosis of the cervical arteries. The mean age of patients who had significant differences between the DUS and CTA was 72.4 ± 12.0 years, while the mean age of patients with the same results between the DUS and CTA tests was 64.3 ± 9.1 years.

A comparison of the degree of stenosis of the internal carotid artery by DUS and CTA (Table 2) showed that out of the patients with less than 50% occlusion on CTA (the gold standard), 90.8% were also identified as having less than 50% stenosis by DUS. Among the patients with stenosis of 70% or more on CTA, 73.3% showed the same result on the DUS. Out of the patients with 50–69% stenosis found by CTA, 45.5% of them had different percentage of stenosis by DUS. An example of different results for a single patient shown by DUS and CTA can be seen in Figure 1 (presenting DUS results) and Figure 2 (presenting CTA results).

Comparing the degree of stenosis of the vertebral artery by DUS and CTA (Table 3) while considering CTA as the gold standard showed that DUS was a specific (94.7%) but not a sensitive modality for identifying near complete occlusion. The positive predictive value was 47.1%, and negative predictive value was 95.3%.

Patients with complete occlusion of both the internal carotid and vertebral arteries by DUS had the same results by CTA (Table 4).

## 4. Discussion

In our retrospective study, there were 100 stroke patients who underwent radiological evaluation by both DUS and CTA. The main findings of the current study are the significant differences found in the evaluation of stenosis, in both the internal carotid arteries and vertebral arteries, between DUS and CTA. Differences between the testing methods were most evident in elderly stroke patients. Complex structural and functional changes in the arterial system with atherosclerotic plaque calcification in elderly patients complicate the execution of DUS by the technician and may explain these discrepancies.

In the evaluation of the degree of carotid arteries stenosis, the most significant mismatches were observed in the group of patients who had 50–69% stenosis observed by DUS. Out of these, only 54.5% patients had the same percentage of stenosis by CTA, 18.2% had overestimated results by DUS and less than 50% stenosis by CTA, and 27.3% patients revealed underestimated results by DUS and 70–98% stenosis by CTA. 

Importantly, the evaluation of the degree of vertebral artery stenosis by DUS and CTA showed correlation in just 47.1% of the patients with more than 50% stenosis by DUS. However, 52.9% of the patients with abnormal DUS had normal or less than 50% stenosis by CTA. Overall, all patients with complete occlusion of the internal carotid or vertebral artery by DUS had the same results by CTA. 

In our study we also looked for the clinical manifestation of stroke in patients, and the severity of stroke. We tested risk factors including diabetes, hypertension, hyperlipidemia, and a history of smoking, as shown in Table 1. However, none of these factors affected the differences between the DUS and CTA evaluations. 

While the absence or low degree of stenosis could be accurately detected using DUS, the modality less reliably detected moderate to severe occlusion. According to our findings, we recommend that patients who show stenosis of carotid arteries of 50% or more by DUS should also be evaluated by CTA.

To our knowledge, this is the first study to compare the DUS and CTA results in cervical artery stenosis in stroke patients. Previous studies have compared between DUS and CTA, but focused mainly on the evaluation of carotid artery stenosis (CAD). In some of these studies, there was concordance between the DUS and CTA methods. Accordingly, a study published in 2018 by Boyko et al. compared carotid DUS to other angiographic modalities in the measurement of the carotid artery stenosis in 245 patients and showed excellent agreement between DUS and CTA [18].

Other studies have shown that CTA is more precise than DUS in the evaluation of carotid stenosis. Anderson GB studied CTA for the detection and characterization of carotid artery bifurcation in 40 patients who underwent CTA, DUS, and digital subtraction angiography (DSA). This study from 2000 showed that the correlations between DUS and DSA were poorer than those between CTA and DSA [15]. A study by Titi et al. (2007) compared carotid DUS and CTA in the evaluation of carotid artery stenosis in 107 patients. This study showed that the overall concordance between both DUS and CTA was 79.1% [17].

In a study by Rozeman et al. for the detection of vertebral artery stenosis, DUS appeared to be limited in detection compared with CT angiography. In their study, the sensitivity of DUS was 39% and the specificity was 88%, with a corresponding positive predictive value (PPV) of 23% and the negative predictive value (NPV) of 94%. This finding fits our study findings. 

In our study, only 47.1% of patients with more than 50% stenosis in the vertebral artery observed by DUS had the same result with CTA. The remaining 52.9% of patients had normal or less than 50% stenosis by CTA. The sensitivity of DUS was 50% and the specificity was 94%, with a corresponding positive predictive value (PPV) of 47% and the negative predictive value (NPV) of 95%.

DUS has limitations, mainly as it is operator and experience dependent [8], and is also related to the patient’s physical condition (e.g., obesity, heart failure, postoperative status) [22]. Through assessment of the vertebral artery, DUS may have technical difficulties such as the often deep and posterior origin of the vertebral arteries, calcified lesion, torturous course, or short neck stature [19]. Additional disadvantages of the DUS method include limited visualization of the proximal common carotid and distal internal carotid arteries [22]. 

The main strengths of the current study are the evaluation of the anterior and posterior circulation by DUS and CTA at the time of emergency room assessment and in patient admission, along with the evaluation of the degree and site of stenosis, clinical manifestation of stroke, and severity of stroke by NIHSS in a relatively large sample size. Additional information collected included age and risk factors such as diabetes, hypertension, hyperlipidemia, and a history of smoking. 

The main limitations of this study are its retrospective nature and its single-center setting, which may lead to sampling errors and selection bias. Additionally, we lack data on clinical follow-up for patients after discharge. Therefore, the generalization of our results to other geographical areas should be explored in future prospective multi-center studies. 

## 5. Conclusions

In conclusion, the current study confirms that CTA is more accurate than DUS in the evaluation of stenosis of the cervical arteries in both the internal carotid and vertebral arteries. Most significantly, in older stroke patients, there were more obvious differences between DUS and CTA. Specifically, the differences were most abundant in the group of patients who had 50–69% stenosis of an internal carotid artery by DUS or abnormal evaluation of a vertebral artery by DUS. Overall, all patients with complete occlusion of the internal carotid or vertebral artery by DUS had the same results by CTA. 

To date, the international standard of care is the Doppler test, which is performed up to 24 h after stroke. We recommend including the CTA as part of the standard of care of stroke patients as it can detect all degrees of stenosis and achieve a more complete diagnosis, which has implications for medication, the need for other procedures as well as the risk of recurrent strokes.

## Figures and Tables

**Figure 1 diagnostics-13-00459-f001:**
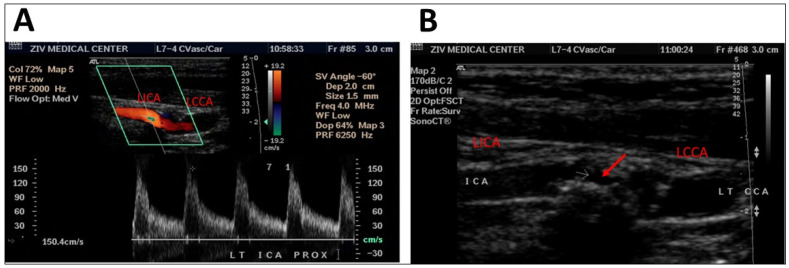
DUS scan of a patient showing moderate stenosis of 50–70%, as seen by an increase in the flow velocity of the left internal carotid artery (**A**) and by occlusion site (**B**, red arrow). LICA = left internal carotid artery, LCCA = left common carotid artery, ICA = internal carotid artery, LT = left, CCA = common carorid artery.

**Figure 2 diagnostics-13-00459-f002:**
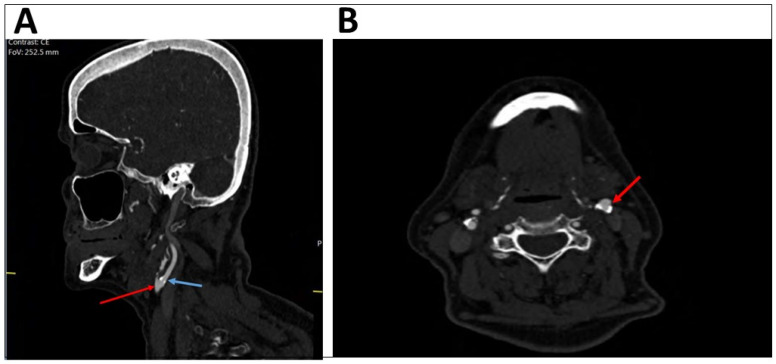
CT angiography showing the same patient with mild stenosis of the left internal carotid artery. The sagittal plan (**A**) presents the left common carotid artery (bulb, red arrow) and the left internal carotid artery (blue arrow). The axial plan (**B**) shows mild stenosis of the left internal carotid artery (red arrow).

**Table 1 diagnostics-13-00459-t001:** Characteristics of the study patients.

Variables	
Age, years (Mean ± SD)	68.8 ± 11.5
Gender (n, %)	72.4 ± 12.0
Male	63, 63.0
Female	37, 37.0
Ethnicity (n, %)	
Jewish	69, 69.0
Other	31, 31.0
Smoking, Yes (n, %)	19, 23.2
Hypertension, Yes (n, %)	82, 82.0
Diabetes mulitas, Yes (n, %)	51, 51.0
Hyperlipidemia, Yes (n, %)	75, 75.0
Origin of stroke (n, %)	
Anterior circulation	38, 59.4
Posterior circulation	26, 40.6
tPA, Yes (n, %)	6, 6.0
NIHSS (n, %)	
0	9, 9.0
1–4	51, 51.0
5–14	38, 38.0
15–20	1, 1.0
20+	1, 1.0

tPA—tissue Plasminogen Activator; NIHSS—National Institutes of Health Stroke Scale.

**Table 2 diagnostics-13-00459-t002:** Comparison of the degree of stenosis of right (RT) and left (LT) internal carotid artery by DUS and CTA.

	Computerized Tomographic Angiography (*n* %)			
Doppler Ultrasound	Normal/<50%	50–69%	70–98%	*p*
Normal/<50%	148, 90.8	12, 7.4	3, 1.8	<0.001
50–69%	2, 18.2	6, 54.5	3, 27.3	
70–98%	1, 6.7	3, 20.0	11, 73.3	

**Table 3 diagnostics-13-00459-t003:** Comparison of the degree of stenosis of the right (RT) and left (LT) vertebral artery by Doppler Ultrasound (DUS) and Computerized Tomographic Angiography (CTA).

	Computerized Tomographic Angiography (n, %)		
Doppler Ultrasound	Normal/<50%	50–98%/Near Occlusion	*p*
Normal	161, 95.3	8, 4.7	<0.001
Abnormal	9, 52.9	8, 47.1	

**Table 4 diagnostics-13-00459-t004:** Comparison of the complete occlusion in the carotid and vertebral arteries by Doppler Ultrasound (DUS) and Computerized Tomographic Angiography (CTA).

(Degree of Stenosis) Complete Occlusion (100%)	Doppler Ultrasound (n, %)	Computerized Tomographic Angiography (n, %)
Right (RT) & Left (LT) Internal Carotid	11, 100%	11, 100%
Right (RT) & Left (LT) Vertebral	14, 100%	14, 100%

## Data Availability

Full data are available following a formal request and in compliance with state regulations.

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
