# Peer review of "Comparison of Doppler Ultrasound and Computerized Tomographic Angiography in Evaluation of Cervical Arteries Stenosis in Stroke Patients, a Retrospective Single-Center Study"

_diagnostics, 2023, doi:10.3390/diagnostics13030459_

Round 1

Reviewer 1 Report

This is an interesting paper that can have clinical impact. I believe that although there have been similar papers comparing US and CTA for evaluation of atherosclerosis of the carotid arteries, the evaluation of the vertebral arteries is novel. 

I think that shortening the introduction and may be transferring some of the information to the Discussion will make the flow easier to follow. I suggest adding some explanation about the potential role of MRA to evaluate atherosclerosis of the cervical arteries, and how it compares to CTA.

Some of the sentences, especially in the introduction need to be revise to make the text more understandable. 

Author Response

We thank the reviewers for their thoughtful comments which enabled us to revise our manuscript to be more precise and include all necessary details. To this end, all the points raised by the reviewers have been revised accordingly. More specifically:

Reviewer #1

  1. I think that shortening the introduction and may be transferring some of the information to the Discussion will make the flow easier to follow. I suggest adding some explanation about the potential role of MRA to evaluate atherosclerosis of the cervical arteries, and how it compares to CTA.

The introduction has been shortened and a paragraph explaining MRA role, its advantages and limitations comparatively to CTA, has been added.

  1. Some of the sentences, especially in the introduction need to be revise to make the text more understandable. 

The introduction has been totally revised.

Reviewer 2 Report

The manuscript is very interesting, with the authors’ aim to evaluate the concordance between DUS and CTA in evaluation the degree of craniocervical arteries (both carotid and vertebral arteries). The main strength of the manuscript is the evaluation of the vertebral arteries stenosis (which is not easy to perform with DUS), and the number of included patients

However some major issues emerged after reading the manuscript.

MAJOR ISSUES

Title: in the title it should be specified that this is a single center study.

Of the 737 patients with acute ischemic stroke, how were 100 stroke patients selected and included in the study?

The major issues is the following: who performed the DUS in all patients? A single radiologist/neurologist? Two radiologists/neurologists? How were the discrepancies solved? Were DUS evaluated offline? How many days elapsed between the DUS and the CTA? Were radiologist/neurologist performing DUS blinded to the results of the CTA? How was stenosis determined (ECST or NASCET criteria)? And similarly, who performed the CTA? The same radiologist for all 100 stroke patients? Two radiologists? How were the discrepancies solved? The intraclass agreement (intra-DUS and intra-CTA) was evaluated? Were the radiologist performing CTA blinded to the results of DUS?

Table 2: what is the meaning of the asterisk (*)? In this table it should be reported the number of patients with DUS Normal - 50-69% - 79-98%, while it is reported only the number of the three categories for CTA.

Table 3: what is the meaning of the asterisk (*)? In this table it should be reported the number of patients with DUS Normal - Abnormal, while it is reported only the number of the two categories for CTA.

Why the authors found that “differences between testing methods were most evident in elderly age stroke patients”? An hypothesis should be formulated (such as: more difficult execution of DUS due to cervical ossification).

The authors said that “We tested risk factors including diabetes, hypertension, hyperlipidemia and a history of smoking.”. Why these data were not shown in the results section, but discussed in the discussion section?

MINOR ISSUES

The sentence: “Most stroke prevention treatments (e.g., cholesterol-lowering agents and antiplatelet therapy) are unlikely to have immediate effect or may be unsafe if implemented too quickly (e.g., anticoagulant and antihypertensive agents)” is repeated twice.

Correct “collateral channels” with “collateral vessels”.

Correct the capital letter “no enhanced CT [16-17]. computerized”.

Table 1: adjust alignment of the cells. Explain in the footnote the meaning of “with no difference / with difference”.

Author Response

We thank the reviewers for their thoughtful comments which enabled us to revise our manuscript to be more precise and include all necessary details. To this end, all the points raised by the reviewers have been revised accordingly. More specifically:

Reviewer #2

Major issues

  1. Title: in the title it should be specified that this is a single center study.

The title has been modified as suggested.

  1. Of the 737 patients with acute ischemic stroke, how were 100 stroke patients selected and included in the study?

Of the 737 patients admitted to our department with an acute ischemic stroke, only the patients who underwent radiological evaluation with both modalities DUS and CTA were selected and included in this study.

  1. The major issues is the following: who performed the DUS in all patients? A single radiologist/neurologist? Two radiologists/neurologists? How were the discrepancies solved? Were DUS evaluated offline? How many days elapsed between the DUS and the CTA? Were radiologist/neurologist performing DUS blinded to the results of the CTA? How was stenosis determined (ECST or NASCET criteria)? And similarly, who performed the CTA? The same radiologist for all 100 stroke patients? Two radiologists? How were the discrepancies solved? The intraclass agreement (intra-DUS and intra-CTA) was evaluated? Was the radiologist performing CTA blinded to the results of DUS?

The same US technician performed the DUS in all the patients and was blinded to the CTA results. The CTA were performed by the same neuroradiologist who was blinded to DUS evaluations. 72 hours maximum elapsed between DUS and CTA assessments. Stenosis was determined according to NASCET criteria. The question concerning the intraclass correlation is not relevant in our study given that the same US technician and the same neuroradiologist performed respectively the assessments.      

  1. Table 2: what is the meaning of the asterisk (*)? In this table it should be reported the number of patients with DUS Normal - 50-69% - 79-98%, while it is reported only the number of the three categories for CTA.

  1. Table 3: what is the meaning of the asterisk (*)? In this table it should be reported the number of patients with DUS Normal - Abnormal, while it is reported only the number of the two categories for CTA.

The asterisks appeared by mistake in both tables. The tables have been corrected, number of patients (n) and percentage (%) have been added for more clarity. Each result represents a combination of the frequency and the percentage for both modalities.

  1. Why the authors found that “differences between testing methods were most evident in elderly age stroke patients”? A hypothesis should be formulated (such as: more difficult execution of DUS due to cervical ossification).

Complex structural and functional changes in the arterial system with atherosclerotic plaques calcification in elderly patients complicate the execution of DUS by the technician and may explain these discrepancies.

  1. The authors said that “We tested risk factors including diabetes, hypertension, hyperlipidemia and a history of smoking.”. Why these data were not shown in the results section, but discussed in the discussion section?

These data are shown in Table 1.

Minor issues

  1. The sentence: “Most stroke prevention treatments (e.g., cholesterol-lowering agents and antiplatelet therapy) are unlikely to have immediate effect or may be unsafe if implemented too quickly (e.g., anticoagulant and antihypertensive agents)” is repeated twice.

This duplicate sentence has been removed.

  1. Correct “collateral channels” with “collateral vessels”.

The paragraph with this sentence has been removed from the introduction.

  1. Correct the capital letter “no enhanced CT [16-17]. computerized”.

The paragraph with this sentence has been removed from the introduction.

  1. Table 1: adjust alignment of the cells. Explain in the footnote the meaning of “with no difference / with difference”.

These data, not appropriate in Table 1, have been removed and explained in the results.

Round 2

Reviewer 1 Report

The authors have appropriately addressed my initial concerns. 

Reviewer 2 Report

Authors answered to all my previous comments, and now the manuscript has improved. I have no further comments.